# Prevalence of Work-Related Musculoskeletal Disorders and Its Effects amongst Nurses in the Selected Intellectual Disability Unit of the Limpopo Province

**DOI:** 10.3390/healthcare11050777

**Published:** 2023-03-06

**Authors:** Livhuwani Muthelo, Nakisani Faith Sinyegwe, Thabo Arthur Phukubye, Masenyani Oupa Mbombi, Tshepo Albert Ntho, Tebogo Maria Mothiba

**Affiliations:** 1Department of Nursing, University of Limpopo, Private Bag X1106, Sovenga, Polokwane 0727, South Africa; 2Faculty of Health Science Executive Dean’s Office, University of Limpopo, Private Bag X1106, Sovenga, Polokwane 0727, South Africa

**Keywords:** prevalence, effects, work-related musculoskeletal disorders, intellectual disability, nurses

## Abstract

The COVID-19 pandemic continuously highlights the need for occupational health and safety among healthcare professionals. Physical and mental health safety resulting from needle prick injuries, stress, infections, and chemical hazards are priority work-related musculoskeletal disorders for nurses, including those working in the intellectual disability unit. The intellectual disability unit provides basic nursing care to patients with known mental disabilities, such as learning, problem-solving, and judgment problems, which demand diverse physical activities. Nevertheless, the safety of nurses within the unit receives little attention. Thus, we adopted a quantitative cross-sectional epidemiological survey design to determine the prevalence of work-related musculoskeletal disorders amongst nurses working in the intellectual disability unit at the selected hospital in Limpopo Province, South Africa. A self-administered questionnaire collected data from 69 randomly selected nurses from the intellectual disability unit. Data were extracted, coded, and captured in MS Excel format (2016) and imported into the IBM Statistical Package for the Social Sciences (SPSS), software version 25.0, for analysis. The study reported a low (38%) prevalence of musculoskeletal disorders in the intellectual disability unit, with significant effects on nursing care and staffing. The effects of these WMSDs included missing work, interference with the daily routine, disturbance with sleeping patterns after work, and absenteeism from work. Since intellectually disabled patients depend entirely on nurses for the basic activities of daily living, this paper recommends the incorporation of physiotherapy among nurses in the intellectual disability unit to treat the incidence of lower back pain while mitigating nurses missing work or absenteeism.

## 1. Introduction

The COVID-19 pandemic paved the way for more deliberations on the significance of occupational health and safety issues among nurses and other healthcare professionals. According to WHO, yearly, over two million people worldwide die of occupational injuries and work-related musculoskeletal disorders [1]. Work Musculoskeletal Disorders (WMSDs) are a group of painful disorders of muscles, tendons, and nerves affecting the wrists, elbows, neck and shoulders, hands, legs, hips, ankles, and feet, such as carpal tunnel syndrome, tendonitis, thoracic outlet syndrome, and tension neck syndrome [2].

Globally, the prevalence of WMSDs is high in different continents. For instance, in America, WMSDs range from 35.1% to 47%; in Asia, WMSDs range from 78.6% to 88% [3,4,5]. In Africa, the prevalence of WMSDs ranges from 44.1% to 94% [6,7]. Worldwide, WMSDs are a common cause of morbidity affecting, among others, healthcare professionals. WMSDs impact nurses, healthcare facilities, and the national economy differently. For instance, WMSDs cost substantial expenditures on nurses’ medical care and compensation costs annually, as seen in the United States of America, where musculoskeletal disorders are responsible for an economic burden of about USD 45 and 54 billion annually [8,9]. Furthermore, WMSDs account for nearly 70 million physician office consultations annually, with an estimated 130 million healthcare encounters. Musculoskeletal disorders present a severe public health problem affecting work performance with various personal, social, and economic impacts on nursing. For example, an increased sick leave and absenteeism among nurses due to WMSDs impacting their quality of employment life has been reported [10]. Equally, Chang et al. reported that WMSDs among nurses threaten their quality of life [11]. Therefore, WMSDs are a serious workforce challenge that exacerbates early retirement and absenteeism among nurses [12]. To minimize the overhead costs, it is significant that the prevalence of WMSDs and prioritizing the intervention be determined.

In addition, Albanesi, Piredda, Bravi, Bressi, Gualandi et al. reported on the need for more efforts to extend the multifaceted interventions of WMSDs among nurses [13]. So far, the most reported interventions target WMSDs related to needle prick injuries, physical injuries, stress and impact on mental well-being, infections, chemical hazards, radiation exposure, and work-related exhaustion, which are dominant occupational hazards experienced by nurses during the execution of nursing care duties in selected healthcare units [14,15]. Amare et al. noted that WMSDs occur in various healthcare units, including intellectual disability units. The current paper focuses on Work-related Musculoskeletal Disorders (WMSDs) experienced by nurses in an intellectual disability unit pre-COVID-19 pandemic. Intellectual disability units care for patients with known mental disabilities, such as learning, problem-solving, and judgment, which demand diverse physical activities [15]. Intellectual disability means a significantly reduced ability to understand new or complex information, learn and apply new skills, and a reduced ability to cope independently [16,17]. Some nursing activities rendered in the intellectual disability unit include meeting basic needs (personal hygiene, nutritional and safety needs), position changing, and intellectual support, achieved in collaboration with a multidisciplinary team [18,19]. The required nursing care in the intellectual disability unit poses a high risk of developing WMSDs [20,21], but very little is known about the prevalence of WMSDs in this unit.

WMSDs are common in the healthcare sector, with prevalence rates varying from 28% to 96% over one year, including among nurses [22]. According to reported data worldwide, nurses have a very high prevalence of WMSDs. For instance, in Africa, Uganda reported an 80.8% prevalence of WMSDs among nurses [23] as the burden associated with lifting patients, heavy load, patients transfer from the floor and out of bed, and working in awkward positions [10]. The study conducted in Zimbabwe by Chiwaridzo, Makotore, Dambi, and Munambah [24] revealed that 82% of nurses working in maternity, surgical, and medical units experienced WMSD. Conversely, in Nigeria, 84.4% of the nurses were found to have had WMSDs or more during their years of work [25,26]. In South Africa, a study conducted in the healthcare facilities of KwaZulu-Natal indicated that nurses are more prone to injuries since their duties include manual lifting of patients and bending, a significant cause of WMSDs amongst nurses [27].

Several studies have investigated the prevalence of WMSDs amongst nurses in the healthcare sector and the factors behind or the cause. These disorders occur in various body parts, including the neck, shoulder, arm, wrist, and lower back, where back pain is the most prevalent [12,17]. Low back pain is the most common WMSD among nurses, with an estimated 200,000 nurses suffering each year, which costs the National Health Services approximately GBP 45 million [27,28], equivalent to ZAR 87 391,395,000 billion in South Africa. These low back pains and other WMSDs are primarily experienced in various nursing care areas: during patient care, bathing patients, bed making, cleaning and dressing wounds, and medication administration [15]. Considering the nature of the work, nursing is one of the occupations in a healthcare environment where WMSDs are highly prevalent and the most difficult [19,24]. Several factors leading to musculoskeletal disorders among nurses include the frequent lifting of heavy objects, moving the patients, inappropriate physical posture, a fixed and constant posture of the neck, longstanding, excessive rotations, psychological factors, work experience, Body Mass Index (BMI), increasing working hours, and increasing age as risks to developing WMSDs [13,19,20,29,30]. These causes of WMSDs have been observed among nurses working in surgery units (17.8%), emergency (15.6%), Out-Patient Departments (77%), and intensive care units (12.6%), indicating a gap for further studies in other units, such as the intellectual disability unit [31].

Subsequently, providing nursing care to patients with an intellectual developmental disorder is a universal concern to all nurses due to occupational health hazards, including WMSDs impacting the nursing care provided [32]. Although various studies have been conducted on the causes of WMSDs, less is deliberated regarding the prevalence and effects of WMSDs among nurses and their working conditions in the intellectual disability units. The current study aims to determine the prevalence of work-related musculoskeletal disorders and their effects amongst nurses working in the selected intellectual disability unit of Limpopo Province. The study findings will contribute to the body of knowledge for achieving the objective of the South African Department of Health National Strategy, which aims to protect the health and safety of healthcare workers while ensuring mental health.

## 2. Materials and Methods

### 2.1. Study Site

A quantitative cross-sectional design enabled the authors to obtain the prevalence data and the effects of WMSDs on nurses. The study was conducted in a psychiatric selected hospital providing services for patients with an intellectual disability in Collins Chabane Local Municipality, Vhembe district of Limpopo Province. This selected hospital is the only psychiatric hospital serving semi-urban and deep rural communities in the Vhembe district. The hospital has ten nursing care units with a 390-bed capacity, and the intellectual disability unit is the focus of the study.

### 2.2. Population and Sampling

The target population in this study was all registered nurses, enrolled nurses, and auxiliary nurses working in both male and female intellectual disability units. A non-probability purposive sampling technique was used to select the nurses in the intellectual disability unit with more than six months of working experience. In addition, to enhance accessibility to more nurses, the study included only nurses working day shifts. This paper excluded nurses who are not working in the intellectual disability unit and those that have less than six months of working in the unit. Using non-probability purposive sampling allowed for the deliberate identification and selection of individuals with characteristics matching the phenomenon of interest [33]. The target population for the study was 90, and a reliable sample size of 73, with a confidence of 95%, and a margin of error of 5%, was established using the Raosoft sample calculator.

### 2.3. Data Collection

The primary author approached the intellectual disability unit nurse manager after obtaining ethical clearance from Turfloop Research Ethics Committee (TREC/160/2018: PG). Furthermore, the Limpopo provincial Department of Health and the hospital CEO granted permission to approach nurses. The nurse manager assisted in arranging the safe cubicle for data collection and recruitment of nurses for participation in the study. The primary author collected data for two months (September to October 2018) using a self-administered questionnaire. A total of 73 self-administered English-written questionnaires consisting of four sections were distributed, with a response rate of 94.5%. Four questionnaires were spoiled, and a total of 69 questionnaires were analyzed. The four questionnaire sections included Section A: Demographic data (gender, age, working experience, weight, and previous history of WMSD); Section B: (discomfort, strains, sprains, or tears; impairment, disability, persistent pain, and carpal tunnel syndrome); Section C (general health and history of treatment for WMSD); and Section D (incidence of WMSD).

### 2.4. Data Analysis

Data analysis entails contrasting and comparing the final data to determine which patterns and themes emerge [34]. As assisted by a faculty biostatistician, descriptive statistics were applied to analyze 69 questionnaires after four were spoiled. Data were extracted, coded, and captured in MS Excel format (2010) and imported into the IBM Statistical Package for the Social Sciences (SPSS), software version 25.0, for analysis. Data are presented in the tables and graphs by percentages and frequency distribution.

### 2.5. Reliability and Validity of the Study

Reliability was ensured by conducting a small-scale study before the main study from 10% of the target population to determine the suitability of the questionnaire and if the time allocated was adequate. The small-scale study resulted in no changes to the main data collection tool.

Content validity was ensured by giving the questionnaire to the last two authors who specialized in the field for review and recommendations. At the same time, face validity was ensured by doing a literature review and providing a questionnaire to the researcher with an expertise in quantitative research [35].

## 3. Presentation of Results

### Demographic Profile of the Participants

The demographic profile is presented according to gender, age weight, and working experiences of nurses in the intellectual disability unit.

As depicted in Figure 1 above, most of the nurses in this study were female, 75%, with males making up 25%. This indicates that the intellectual disability wards were dominated by females, which might be because the nursing profession is predominantly a female-dominant profession in South Africa. Regarding age, the study had respondents between 41 to 50 years, the highest with 38%, followed by those aged 31–40 (29%). This indicates that most of the nurses in the intellectual disability unit were middle-aged and probably more experienced.

As indicated in Figure 2, the majority (44%) of nurses in the intellectual disability unit had between 1 and 10 years of working experience, while nurses with more than 30 years of working experience were less (4%). Regarding weight, most respondents weighed 70–80 kg (52%) as compared to 50–60 kg, which is 29%, 90–100, which makes (17%) and 100+ (2%). Physiologically, people weighing 70–80 kg are more prone to developing WMSDs. To support the statement above, Figure 3 below illustrates the prevalence of WMSDs per body part.

The study findings further indicate that lower back pain (43%) is the most experienced WMSDs as compared to the shoulder (22%), hands (12%), lower limbs (10%), neck (9%), and wrist (4%) body parts.

As depicted in Table 1, the nurses working in the intellectual disability unit experience pain (38%) and discomfort strains, sprains, and tears (32%). The respondents aged between 50 and above predominantly suffer from pains aggravated by work 50%; they also experience pain after lifting patients 37.5%, experience pain after longstanding 12.5%, and their work is limited due to pain 25%. Therefore, it assumed that respondents aged 50 and above years are more prone to WMSDs than other age groups.

As illustrated in Table 2, WMSDs significantly affect nurses differently. For example, most nurses (83%) reported missing work, 58%, with almost half of the nurses remaining at work, indicating that it does not interfere with their daily work routine (49%). It presumed that the prevalence of low back ache contributes to most nurses (58%) opting to change their working station due to WMSD. Additionally, we noted that most females (79%) absented themselves from work due to pain compared to 21% of males. It can only be assumed that females are more affected by WMSDs than males. To better understand the association of WMSD, Table 3 below indicates the significant association between WMSDs and the risk factors. Table 3 below illustrates the effects of demographic and work hours per day, working in the same position for more than two hours, manual lifting, and availability of patient lifting, extra job, and not enough rest during the day with WMSDs.

Table 3 above indicates that older people (50+) experienced WMSDs compared to younger people. Most were females (69%) who experienced WMSDs compared to males. Of respondents with experience of 11–20 years, 42% were seen to be more prone to WMSDs than others. Of those allocated for 1–2 years in the same unit, 46% were more affected by WMSD. Almost 92% of respondents are working 12 h per day which is more than recommended. About 81% of respondents manually lift patients and work extra jobs to compensate for the shortage, whereas 54% reported insufficient resting time during the day.

## 4. Discussion of Research Results

The study aimed to determine the prevalence of WMSDs and their effect on nurses working in the intellectual disability unit. Notably, to enhance the reader’s understanding of the study findings, the respondents’ demographic data is the point of departure for the results. Most of the respondents in the study were females (75%) as compared to (25%) of the male participants, which is consistent with various studies [20,26,36,37] attributed to female dominance within the nursing profession. There was no statistical association between variable gender and WMDs in this study. In contrast, other studies reported female health professionals having a 1.9 times higher risk of developing WMSDs [36,38]. Most of the respondents in this study (38%) were between the ages of 41 and 50, followed by those aged 31–40 (29%). This indicates that most of the nurses in the intellectual disability unit where data were collected were middle-aged and probably more experienced. Age was significantly associated with WMSDs in this study (*p*–value = 0.0130 *). Most of the respondents affected were above 50 years of age, followed by those between 31 and 50. The younger generation, between 20–30, had no problems with WMDs; this could be because their functional capabilities decrease when people age. The findings of this study concern the study of Anap, Iyer, and Rao, where the older generation of employees was more affected by WMSDs than the younger age group, and one could attribute it to long years of exposure to repetitive activities [39]. The study by Sethi, Sandhu, and Imbanat indicates that workers with high Body Mass Index (BMI) were found to be at risk for WMSDs and occupational psychosocial stress because overweight could be the factor to contribute to increasing the physiological and mechanical load on tissues [40]. In this study, most respondents weighed 70–80 kg (52%), as compared to 50–60 kg, which is 29%, 90–100, which makes (17%) and 100+ (1%).

The current study looked at weight but not BMI, and so it can therefore be assumed that respondents of weight 70–80 and more are prone to develop WMSDs. There was no significant association between years of working experience between WMSDs with the years of working experience. This was the same in the study of Akodu and Ashalejo, where the variable age (*p* = 0.447), gender (*p* = 0.801), and years of working experience (*p* = 0.873) had no significant association with WMDs [26]. Karkousha and Elhafeza in Egypt also revealed no significant association between WMSDs and years of working experience [41]. A Chi-square test was employed to determine whether there is a significant association between the WMSD and risk factors. It was found that the *p*-value for gender, years of experience, years allocated in the unit, hours of work per day, working the same position for more than two hours, manual lifting, working an extra job to compensate for shortage, and not enough resting during the day was more than the significance level (0.05). Thus, we conclude that there is no relationship between WMSDs and gender, years of experience, years allocated in the unit, hours of work per day, working the same position for more than two hours, manual lifting, working an extra job to compensate shortage, and not enough resting during the day. However, age and availability of patient lifting variables were statistically associated with WMSD.

The study also determined the prevalence of WMSDs in terms of persistent pain (38%) compared to the 32% of nurses who experienced discomfort, sprains, or tears. The study findings are congruent with the results of Tariah, Nafai, Alajmi, Almutairi, and Alanazi and also in the study by Rypicz, Karniej, Witczak, and Kołcz where most of the respondents (63.8% and 85%), respectively, indicated that they experienced pains during the last 12 months, particularly in the lower back, followed by shoulders, 50% and 67%, respectively [30,42]. Contrary to this study’s findings, in terms of the original body pain, shoulder pain was the highest at 85%, and lower back pain was the lowest at 60.4% in the results of Lin, Lin, Liu Fang, and Liu in Taiwan [43]. Despite the differences in experiencing pain in a body location, nurses in the intellectual disability unit reported persistent pain like others in the surgical unit, intensive care units, and medical ward [24,31]. Therefore, there is a need to strictly monitor the health conditions in the healthcare sectors to rule out the WMSDs and enhance quality healthcare.

The study again determined the association between WMSDs and associated risk factors, with age and patient lifting, indicating a significant association with the disorders (*p* < 0.05). The findings demonstrate that most nurses between 41 and 50 years are at risk for developing WMSDs when lifting patients in the intellectual disability unit. Krishnan, Raju, and Shawkataly also highlighted various physical factors that can impact nurses’ health in a hospital setting [29]. Lifting patients in this study is also a risk factor associated with WMSDs among nurses working in the intellectual disability unit. The physical demands that nurses perceive typically include moving objects, pulling/pushing machines, lifting patients, repeating motions, and sudden or extreme flexion, bending, and twisting. Additionally, Heidari et al. and Saberioura et al. pinpointed factors leading to musculoskeletal disorders in nurses, such as the frequent lifting of heavy objects, moving the patients, inappropriate physical posture, a fixed and constant posture of the neck, longstanding, excessive rotations, psychological factors, work experience, increasing working hours, and increasing age as risks to developing WMSDs [19,20]. Nursing rehabilitation patients are particularly demanding because they must be moved around for multiple activities throughout the day, and nurses frequently do not have enough time to accommodate these patients’ hectic schedules [44]. Teeple, Collins, Shrestha, Dennerlein, Losina, and Katz recommended a safe patient lifting and mobilization program as an effective intervention for WMSDs [45]. Additionally, Lee and Rempel reported a low prevalence of back pain among nurses using ceiling lifts following lifting patients. Therefore, to minimize the risk of back pains from patient lifting among nurses in the intellectual disability unit, ceiling lifts and safe-patient lifting programs are recommended [46]. Furthermore, the findings suggest a need to capacitate the nursing staff to enhance sharing of the nursing responsibility of patient lifting during nurse care of patients in the intellectual disability unit.

WMSDs were again discovered to be the primary reasons for absenteeism, necessitating a change in the nursing shifts. This is consistent with the findings from other scholars who noted nursing staff missing work due to WMSDs [12,29,30]. Additionally, 49% of the female respondents said they had requested a change in duty, compared to only 9% of the male respondents. This suggests that WMSDs affect females more than males in this study, and given the fact that nursing is a female-dominated profession, these results can negatively impact patient care and the healthcare system in general, as most of the nurses might enter into early retirement due to WMSDs. Furthermore, the effects of WMSDs perpetuate absenteeism among nurses working in the intellectual disability unit and negatively impact the nursing care of patients. For example, a study conducted in a tertiary hospital in Limpopo Province reported the effects of absenteeism on nurses remaining on duty. These effects include psychological stress, low morale, and increased workload among nurses remaining on duty [47].

Intellectual disabled patients depend entirely on nurses for the basic activities of daily living, and if more nurses absent themselves due to WMSD, then it is a challenge to these vulnerable groups as they depend entirely on nurses. Policymakers should develop some measures or strategies that mitigate the impact of WMSD, especially within the intellectual disability units, such as training on injury prevention.

### Limitation of the Study

Since the study was conducted only with nurses working in the intellectual disability units of one hospital, this contributed to a small sample. Therefore, the results cannot be generalized to other hospitals in the Vhembe district or outside the province. A literature integration was done to confirm the findings to compensate for the smaller sample size.

## 5. Conclusions

The study reported a low (38%) prevalence of musculoskeletal disorders in the intellectual disability unit, with significant effects on nursing care and staffing. The effects of these WMSDs included missing work, interference with the daily routine, disturbance with sleeping patterns after work, and absenteeism from work. Since intellectually disabled patients depend entirely on nurses for the basic activities of daily living, this paper recommends the incorporation of physiotherapy among nurses in the intellectual disability unit to treat the incidence of lower back pain while mitigating nurses missing work or absenteeism.

## Figures and Tables

**Figure 1 healthcare-11-00777-f001:**
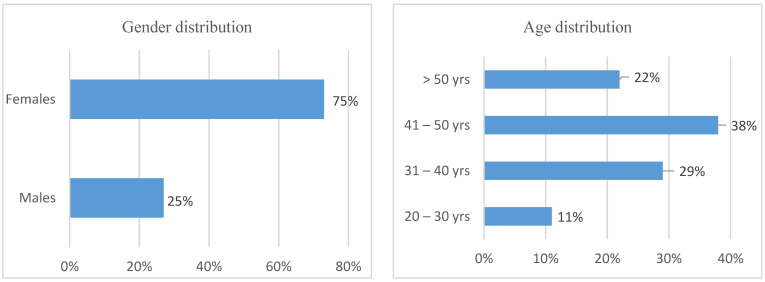
Gender and age distribution.

**Figure 2 healthcare-11-00777-f002:**
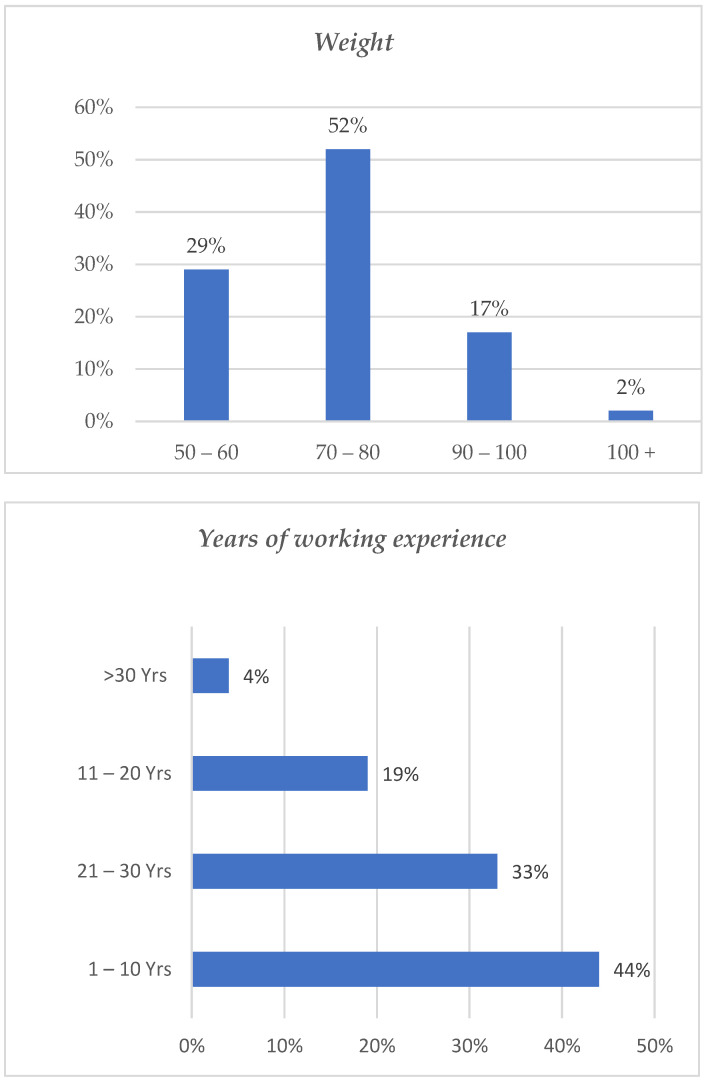
Weight and working experience of nurses.

**Figure 3 healthcare-11-00777-f003:**
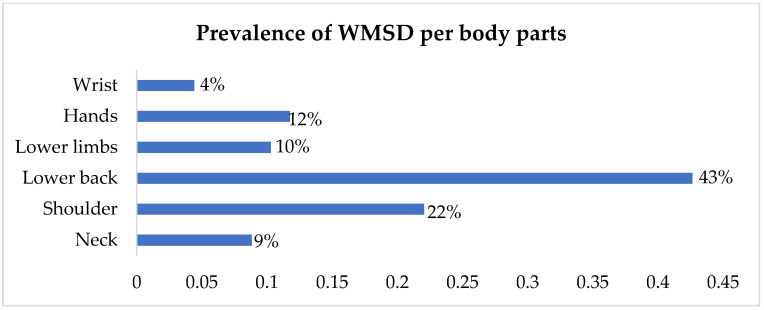
Prevalence of WMSDs per body parts.

**Table 1 healthcare-11-00777-t001:** Incidence of WMSDs per age category.

WMSD	Age Category			
Pain	20–30	31–40	41–50	50+
Pains aggravated by work	0 (0%)	1 (12.5%)	2 (25%)	4 (50%)
Time of working limited due to pain	0 (0%)	2 (25%)	3 (37.5%)	2 (25%)
Experienced pain after lifting patients	0 (0%)	1 (12.5%)	2 (25%)	1 (12.5%)
Experienced pain after longstanding	0 (0%)	4 (50%)	1 (12.5%)	3 (37.5%)
Discomfort, Strains, Sprains, and Tears				
Discomfort, Strains, Sprains, and Tears aggravated by work	1 (5%)	2 (9%)	1 (5%)	3 (14%)
Time of working limited due to Discomfort, Strains, Sprains, and Tears	2 (9%)	0 (0%)	1 (5%)	2 (9%)
Experienced Discomfort, Strains, Sprains, and Tears after lifting patients	1 (5%)	1 (5%)	2 (9%)	1 (5%)
Experienced Discomfort, Strains, Sprains, and Tears after longstanding	1 (5%)	1 (5%)	1 (5%)	1 (5%)

**Table 2 healthcare-11-00777-t002:** Effects of WMSDs among nurses.

Effects	WMSD	
	Yes	No
Absenteeism from work due WMSD	31 (62%)	19 (38%)
Change of working station due to WMSD	31 (62%)	19 (38%)
Missed work due to WMSD	29 (58%)	21 (42%)
WMSD interferes with the daily routine	24 (48%)	26 (52%)
Current work-related WMSD disturbs sleeping patterns	26 (52%)	24 (48%)

**Table 3 healthcare-11-00777-t003:** Association between the WMSD and risk factors.

Risk Factors	WMSD		*p*-Value
	Yes	No	
Age			0.0130
20–30	0 (0%)	8 (19%)	
31–40	8 (31%)	12 (28%)	
41–50	8 (31%)	18 (41%)	
50+	10 (38%)	5 (12%)	
Gender			0.9230
Male	8 (31%)	9 (21%)	
Female	18 (69%)	34 (79%)	
Years of Experience			0.1640
1–10	7 (27%)	23 (44%)	
11–20	11 (42%)	12 (28%)	
21–30	6 (23%)	7 (16%)	
30+	2 (8%)	1 (2%)	
Years in same unit			0.3880
1–2	12 (46%)	27 (63%)	
3–4	9 (35%)	13 (30%)	
5–6	3 (12%)	2 (8%)	
7+	2 (8%)	1 (2%)	
Hours of work per day			0.92300
8	1 (4%)	1 (2%)	
12	23 (92%)	39 (93%)	
12+	1 (4%)	2 (5%)	
Same position more than two hours			
Yes	10 (38%)	14 (33%)	
No	16 (62%)	29 (67%)	
Patient lifting			0.0320
Yes	23 (89%)	28 (65%)	
No	3 (11%)	15 (35%)	
Extra job			0.8650
Yes	21 (81%)	34 (79%)	
No	5 (19%)	9 (21%)	
No enough resting during the day			0.2470
Yes	14 (54%)	17 (40%)	
No	12 (46%)	26 (60%)	

## Data Availability

Due to confidentiality issues, it is not permitted to share the data.

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
