# Peer review of "Prevalence of Work-Related Musculoskeletal Disorders and Its Effects amongst Nurses in the Selected Intellectual Disability Unit of the Limpopo Province"

_healthcare, 2023, doi:10.3390/healthcare11050777_

Round 1
Reviewer 1 Report
Thank you very much for the invitation to review this interesting article on the subject: Prevalence of Work-Related Musculoskeletal Disorders and Its 2 Effects amongst Nurses Working in Intellectual Disability Ward 3 of Hayani Hospital. The interesting quantitative cross-sectional study aimed at identifying and describing the effects of WMSDs amongst the 103 nurses working in the intellectual disability ward at Hayani hospital, Vhembe District in 104 Limpopo.
A few questions and comments:
· The authors should clarify the rationale for conducting the study in the introduction section. In addition, The purpose of the study is still unclear and must be clarified.
· It should be added in the background of the manuscript, what is known about the risk factors for the phenomenon and how they manifest among nurses who work in the ward for intellectual disabilities award.
· Please provide detailed information on the inclusion and exclusion criteria to participate in the study. Also, more information should be provided about the research procedure, how were the nurses approached in practice? How were they recruited? What is known about the health background of the nurses? Do they take painkillers?
· More information should be provided on the reliability and validity of the questionnaire and about the score of the questionnaire and its analysis.
· Details about the ethical committee should be added. Is there IRB number?
· The research conclusions should be relevant to the results of the current study. Please provide them in the abstract and in the discussion's sections.
· The novelty of the present study should be refined in comparison with existing studies.
· There are further limitations such as the sample size which is not necessarily representative of the general population.
Author Response
Thank you for your positive feedback and suggestions. Please find herein the attached

Reviewer 2 Report
The aim of the study was to determine the prevalence of work-related musculoskeletal disorders among nurses working in a ward for people with intellectual disabilities. A self-administered questionnaire was used to collect data from 69 nurses. No information was given on how the questions concerning the occurrence of WMSD were constructed (what was asked specifically, whether about pain or disorders), what period they concerned (ever in life, in the past 12 months, in the last week). In the statistical analysis of the obtained results, only the calculation of the frequency (percentage) and the comparison of distributions were used. The data presented in the tables are inconsistent. Table 1: "Incidence of pain per Age category" shows that pain was present in all respondents, while the number of persons with WMSD was different in subsequent tables. For example, from Table 2: "Effects of WMSD among nurses" we learn that 41 people "Change of working station due to WMSD". However, Table 3: "Association between the WMSD and risk factors" shows that WMSD occurred in 26 persons (8 aged 31-40, 8 aged 41-50 and 10 aged 50+). These differences may be due to the imprecise description of the tables, which must be corrected. In conclusion, I believe that the article in its current form is not suitable for publication.
Author Response
Thank you for your positive feedback and suggestions. Below we have attached responses

Round 2
Reviewer 1 Report
The manuscript is better, but still, the research conclusions should be relevant to the results of the current study. Please provide them in the abstract and in the discussion's sections. Do not mention the result again.
The novelty of the present study should be refined in comparison with existing studies.
Author Response
Thank you for your positive suggestions; the authors have effected corrections. Kindly see the attached.

Reviewer 2 Report
The authors, in accordance with the comments of the reviewers, significantly and satisfactorily improved the first version of the manuscript. I have no other comments and in my opinion the article can be published without further changes.
Author Response
Please find herein the attached
